Citation: *Molecular Systems Biology* 9:684
www.molecularsystemsbiology.com

# Multilevel selection analysis of a microbial social trait

## Laura de Vargas Roditi, Kerry E Boyle and Joao B Xavier*

Program in Computational Biology, Memorial Sloan-Kettering Cancer Center, New York, NY, USA
* Corresponding author. Program in Computational Biology, Memorial Sloan-Kettering Cancer Center, 1275 York Avenue, Box 460, New York, NY 10065, USA.
Tel.: +1 646 888 3195; Fax: +1 646 422 0717; E-mail: xavierj@mskcc.org

The study of microbial communities often leads to arguments for the evolution of cooperation due to group benefits. However, multilevel selection models caution against the uncritical assumption that group benefits will lead to the evolution of cooperation. We analyze a microbial social trait to precisely define the conditions favoring cooperation. We combine the multilevel partition of the Price equation with a laboratory model system: swarming in *Pseudomonas aeruginosa*. We parameterize a population dynamics model using competition experiments where we manipulate expression, and therefore the cost-to-benefit ratio of swarming cooperation. Our analysis shows that multilevel selection can favor costly swarming cooperation because it causes population expansion. However, due to high costs and diminishing returns constitutive cooperation can only be favored by natural selection when relatedness is high. Regulated expression of cooperative genes is a more robust strategy because it provides the benefits of swarming expansion without the high cost or the diminishing returns. Our analysis supports the key prediction that strong group selection does not necessarily mean that microbial cooperation will always emerge.

*Molecular Systems Biology* 9: 684; published online 20 August 2013; doi:10.1038/msb.2013.42
*Subject Categories:* simulation and data analysis; microbiology & pathogens
*Keywords:* conflict; cooperation; metabolic prudence; *Pseudomonas aeruginosa*; swarming

## Introduction

Over the past decade microbiology has shifted perspective to acknowledge that bacteria are not solitary organisms but rather social organisms that rely on a range of population-level traits, such as biofilms, cell–cell communication and cooperative drug resistance (Kolter and Greenberg, 2006; Greenberg, 2010; Carmona-Fontaine and Xavier, 2012). However, often the existence of microbial social traits is justified by their group-level benefits (Costerton *et al*, 1999; Monds and O'Toole, 2009; Lee *et al*, 2010). Social evolution theory predicts that defector phenotypes (i.e., non-cooperative phenotypes, see Table I for our definitions) are favored in mixed populations by individual-level selection (Brannstrom and Dieckmann, 2005; West *et al*, 2006; Nadell *et al*, 2009). In fact, experiments with microbes show that a costly cooperative trait may be favored for its group- or species-level benefits but disfavored in populations where different strains and species mix (Strassmann *et al*, 2000; Griffin *et al*, 2004; Diggle *et al*, 2007; Sathe *et al*, 2010). Understanding how cooperation evolves and remains stable is a key to understanding social traits in bacteria and other microbes (e.g., Crespi, 2001; Fortunato *et al*, 2003; Rainey and Rainey, 2003; Greig and Travisano, 2004; Griffin *et al*, 2004; Fiegna *et al*, 2006; West *et al*, 2006; Diggle *et al*, 2007; Ross-Gillespie *et al*, 2007; Nadell *et al*, 2009; Ross-Gillespie *et al*, 2009; Kummerli *et al*, 2009a, b, c; Sathe *et al*, 2010; Smith *et al*, 2010; Foster, 2011; Koschwanez *et al*, 2011, 2013; Strassmann and Queller, 2011; Xavier, 2011a;

Damore and Gore, 2012; Ratcliff *et al*, 2012; Celiker and Gore, 2012a).

Here, we use a combination of quantitative experiments and mathematical modeling to analyze a model social trait, swarming in *Pseudomonas aeruginosa*, and to determine conditions favoring cooperation. Swarming is a collective form of migration that allows colonies to expand over soft surfaces and thus provides a group benefit. But swarming also requires that individual bacteria secrete massive amounts of rhamnolipid biosurfactants to lubricate the surface (Deziel *et al*, 2003; Caiazza *et al*, 2005). The secreted surfactants are a public good and can be exploited by surfactant-deficient defectors, which benefit from the surfactants secreted by others within the colony without producing surfactants themselves (Xavier *et al*, 2011). In general, if the production of a public good is costly, then defectors can outcompete cooperators within a population and, in the absence of stabilizing processes such as kin selection or discrimination (West *et al*, 2006), eventually drive the cooperative trait to extinction. Nonetheless, many natural isolates of *P. aeruginosa* do secrete rhamnolipids (Deziel *et al*, 1996), which suggests that there are mechanisms favoring and stabilizing rhamnolipid secretion in the wild. A mechanism found recently (Xavier *et al*, 2011) explains that *P. aeruginosa* regulates the expression of the rhamnolipid synthesis operon *rhlAB* using a combination of quorum sensing (the *las/rhl* quorum sensing cascade) and nutrient sensing (Figure 1A). Although not all the

**Table I** Definition of terms used in this paper

| | |
|---|---|
| Cooperative trait | A trait that confers benefits to a recipient individual. The cooperative trait can be costly to the actor (altruistic trait) or beneficial to the actor (mutualism) (West *et al*, 2006) |
| Cooperator | An individual or a strain carrying the cooperative trait, also called the 'actor' |
| Defector | An individual or a strain lacking a cooperative trait but capable of exploiting the cooperation of others |
| Metabolic prudence | A mechanism of gene expression regulation that allows wild-type *Pseudomonas aeruginosa* to produce biosurfactants at no cost to its fitness (Xavier *et al*, 2011) |
| Selfishness | A defector strategy where individuals lack a costly cooperative trait but exploit the cooperation of others |

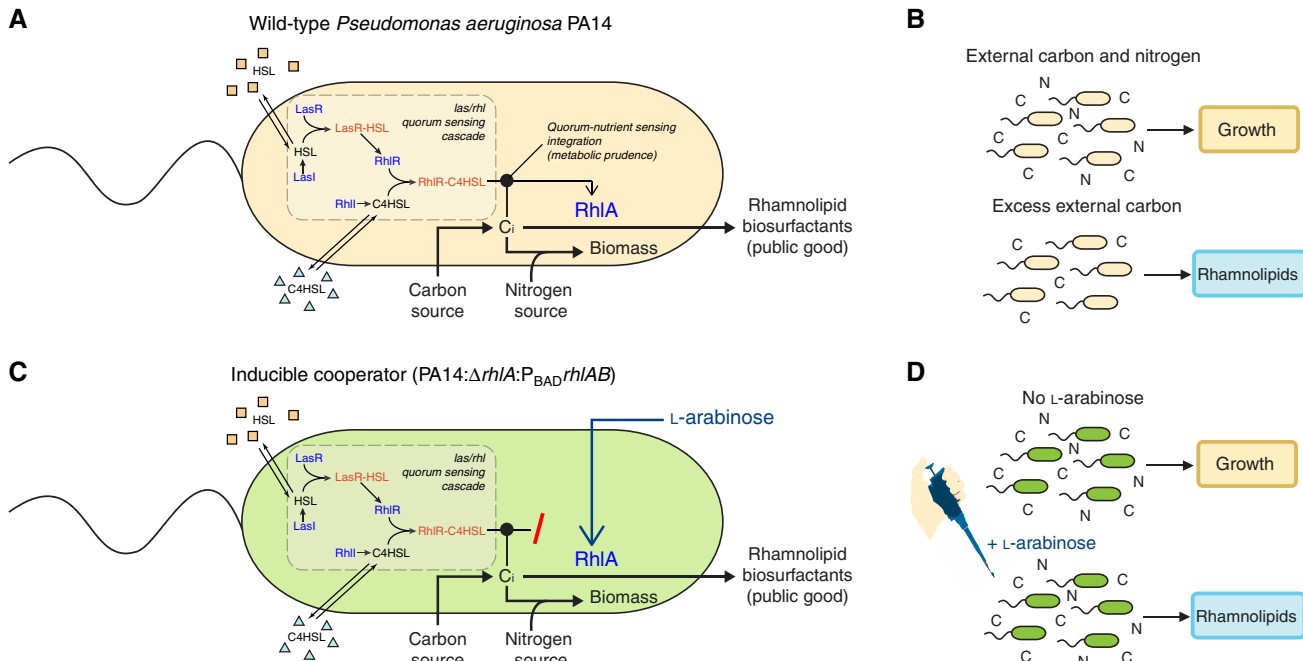

**Figure 1** *Pseudomonas aeruginosa* synthesizes and secretes rhamnolipid biosurfactants required for swarming motility. (**A**) The wild-type *Pseudomonas aeruginosa* regulates the expression of rate-limiting enzyme RhlA using a mechanism called metabolic prudence that integrates nutrient sensing and quorum sensing. (**B**) Metabolic prudence ensures that rhamnolipids are secreted only when carbon is in excess and population density is high. (**C**) In this study, we use a genetically engineered inducible strain where RhlA expression is under regulation of the L-arabinose inducible promoter P_BAD. (**D**) The inducible construct circumvents metabolic prudence and provides an experimental handle to modulate biosurfactant synthesis and investigate swarming cooperation thoroughly.

molecular players in this integration of quorum and nutrient sensing are known (Boyle *et al*, 2013), it is clear that such a combined regulatory mechanism enables bacteria to delay rhamnolipid production to times when there is excess carbon, and rhamnolipid synthesis becomes affordable (Figure 1B). This mechanism, called *metabolic prudence*, implements a molecular decision-making circuitry that effectively decreases the fitness cost of rhamnolipid secretion and prevents exploitation by rhamnolipid-deficient defectors (Xavier *et al*, 2011).

Hamilton's rule (Hamilton, 1964) explains that cooperation can evolve when $br > c$ where $c$ is the fitness *cost* to the actor, $b$ is the fitness *benefit* to the recipient and $r$ is the correlation between the genotypes of actors and recipients, also called *relatedness*. Metabolic prudence reduces the costs of cooperation by regulating the expression of the cooperative genes (Xavier *et al*, 2011), but cooperation could also be favored by increasing the benefits or the relatedness (West *et al*, 2006; Chuang *et al*, 2010). Furthermore, this mechanism brings up an

important question. While some genes are constitutively expressed, many others are conditionally regulated to account for a fluctuating demand for the gene product (Perkins and Swain, 2009); is it better to actively regulate a gene versus constitutively expressing it when the gene regulates a cooperative trait? This question has previously been addressed in other systems (Nadell *et al*, 2008; Kummerli and Brown, 2010; Geisel, 2011) and, for a fixed but arbitrary level, in *P. aeruginosa* swarming (Xavier *et al*, 2011). It remains to be tested whether *P. aeruginosa* metabolic prudence is still a better strategy when constitutive expression of biosurfactants occurs at an optimum rate, and when selection acts at multiple levels.

Here, we investigate multilevel selection in swarming when the expression of cooperative biosurfactant synthesis is kept constant but at a rate that provides optimal group benefits. We use an engineered strain of *P. aeruginosa* that allows us to control the expression of biosurfactant synthesis genes, and thus the investment into cooperation by individuals (Figure 1C

and D). We find that costly swarming can indeed be favored thanks to the large group benefits of population expansion. Furthermore, because swarming effectively expands the population carrying capacity it is more robust than alternative cooperative strategies that improve growth rates but bring only transient benefits to the population. But we also find, as predicted by theory (West *et al*, 2006), that costly swarming cooperation can only evolve under stringent conditions of high genetic relatedness. This is the first time that high relatedness is shown to contribute to the evolution of swarming cooperation and compensate for an unnatural cost non-existing in the wild-type strain. Nonetheless, our results show that strong group benefits alone do not necessarily lead to the evolution of cooperative swarming under multilevel selection, whereas the native regulation by metabolic prudence greatly expands the conditions favoring cooperation.

## Results and discussion

### Swarming is favored by group-level selection but disfavored by individual-level selection

We use an engineered strain of *P. aeruginosa* that has been genetically altered such that the degree of its cooperative effort can be controlled. This strain (PA14 $\Delta rhlA$:$P_{BAD}rhlAB$) has the biosurfactant synthesis genes placed under the regulation of the promoter $P_{BAD}$ so that their expression can be induced by adding L-arabinose to the medium (Boles *et al*, 2005). Previous experiments have shown that inducing biosurfactant with L-arabinose at 0.5% (w/v) has a significant cost in liquid cultures and swarming competitions (Xavier *et al*, 2011). We investigated the induction of swarming cooperation more extensively over a wide range of L-arabinose concentrations (Figure 2A). The results show that inducing surfactant secretion increases fitness of a swarming colony by enabling

spreading over the plate (Figure 2B). However, at high induction levels (L-arabinose > 0.25%) the metabolic costs of surfactant over-secretion start outweighing the benefits and the swarming colonies spread less. The final colony size, and thus the population fitness, peak at intermediate levels of induction (0.25% L-arabinose).

Next, we observed that swarming at 0.25% L-arabinose allows a colony of inducible biosurfactant producers to occupy a larger area on the plate and ultimately to grow better compared with colonies of a defector strain lacking biosurfactant secretion. While the swarming colony reaches the edge of a 9-cm wide Petri dish within 24 h, a colony of surfactant-deficient defectors cannot swarm and the colony stays confined to a region of < 1 cm wide (Figure 2C). The limiting factors affecting bacterial growth on a plate have been extensively studied (Pirt, 1967; Cooper *et al*, 1968; Hochberg and Folkman, 1972). Briefly, the prevailing explanation is that a colony of immotile bacteria, such as our defectors, depletes local nutrients and a nutrient gradient is created as nutrients diffuse toward the colony (Nadell *et al*, 2010). As the region closest to the colony has the lowest concentration of nutrients, the growth of the colony is eventually limited by lack of nutrients. (An alternative mechanism that leads to equivalent outcomes is that toxic waste products accumulate at the immediate surroundings of the colony. Waste products diffuse away from the colony but create a gradient where the concentration is highest closest to the colony where it inhibits growth.) Swarming motility enables a colony to expand beyond the inoculation site, and thus escape the growth-limiting environment.

Comparing the growth of biosurfactant producers (cooperators) and defectors shows that cooperation has a clear benefit in single-strain colonies (Figure 2C). However, natural microbial populations are rarely monoclonal. Processes such as mixing with other strains or species and mutation introduce

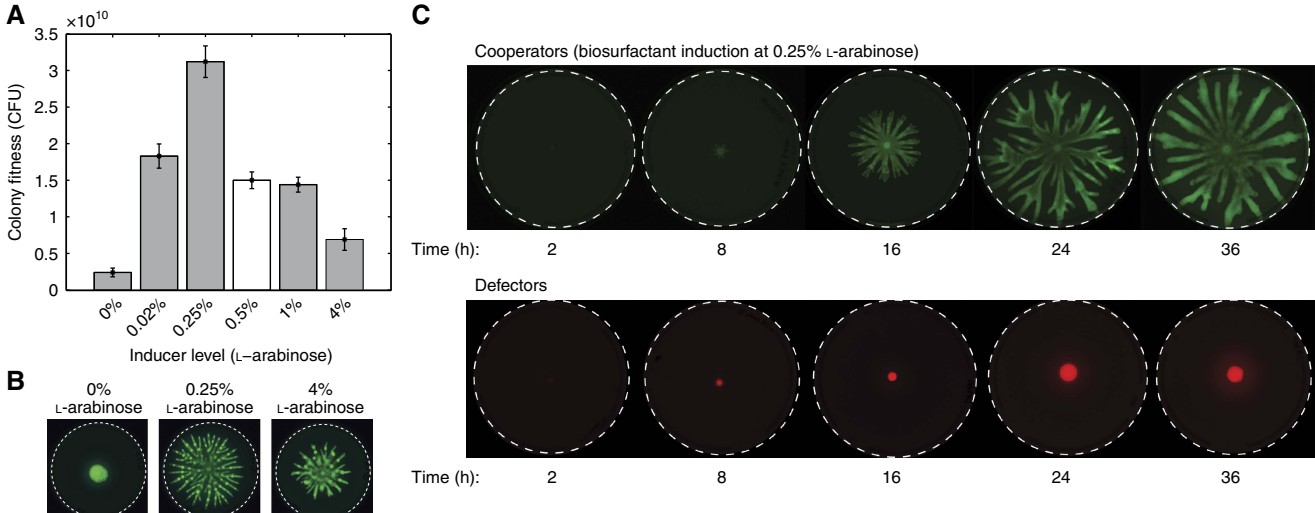

**Figure 2** Cooperative swarming allows *Pseudomonas aeruginosa* colonies to expand over large areas. (**A**) Swarming requires the secretion of biosurfactants. In colonies of the inducible cooperator, fitness peaks at L-arabinose 0.25%, a level at which biosurfactant production is high enough for swarming expansion but low enough that its costs do not overwhelm the benefit of spatial colony expansions. The data point at 0.5% L-arabinose (white bar) comes from a previous study (Xavier, 2011a); all other data (gray bars) were acquired in the present study. CFU = colony forming units. (**B**) Images of swarming colonies. Dashed line represents the edge of Petri dish with 9 cm diameter. (**C**) Cooperators (biosurfactant producers induced by 0.25% L-arabinose) expand over the entire Petri dish. Defectors (rhamnolipid-deficient knockouts) are incapable of swarming and the colonies grow less.

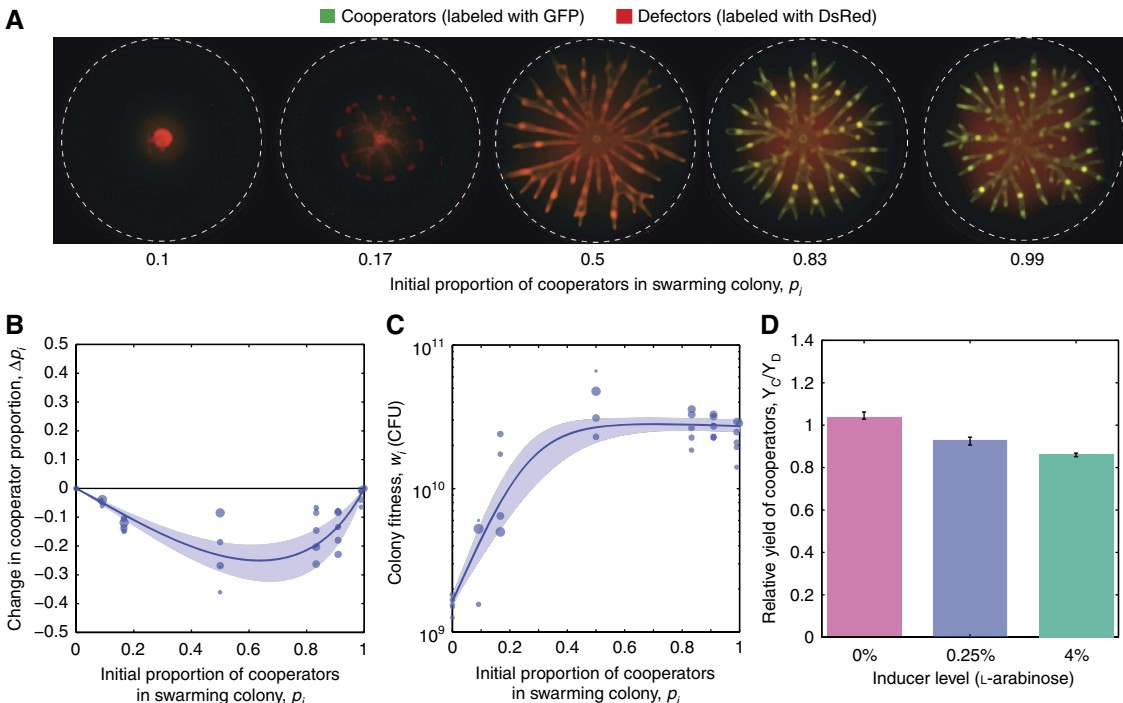

**Figure 3** Individual-level selection disfavors cooperation: induced cooperators increase population size but lose to defectors. (**A**) Competitions between cooperators and defectors mixed at varying mixing proportions (L-arabinose 0.25%). Cooperators are labeled with GFP (green) and defectors are labeled with DsRed-Express (red). (**B**) The change in cooperator proportion due to competition reveals that induced cooperation is strongly selected against. Lines represent best fit for mathematical model (see Materials and methods) and shaded areas represent confidence levels from bootstrapping. Size of data points is proportional to the weight of the data point in the parameter fitting. (**C**) Colony fitness increases with the proportion of cooperators but shows diminishing returns. (**D**) The growth yield of cooperators relative to the yield of defectors decreases with increasing levels of biosurfactant induction. Error bars represent minimum and maximum parameter value obtained from bootstrapping.

genetic variation and influence the evolution of microbial social traits (West *et al*, 2006). Therefore, we investigated whether swarming is still favored when induced cooperators are mixed with defectors within the same colony. For this experiment, we used strains labeled with constitutively expressed fluorescent markers (GFP or DsRed-Express) to allow strain identification (Xavier *et al*, 2011). In each competition, we mixed the two strains at a desired proportion and inoculated approximately one million cells in a 2-µl droplet onto soft agar and incubated for 24 h. The plates were then imaged (Figure 3A) and the final numbers of each strain were determined. We used the data to calculate the changes in cooperator proportion (measuring individual-level selection) and the final colony size (measuring population fitness). As a control, we compared the relative amounts of rhamnolipid secreted in liquid medium by the inducible strain when alone and mixed with 50% defectors to determine whether the presence of defectors affected biosurfactant production. The amount of rhamnose produced by the inducible strain alone is significantly different compared with the strain mixed with 50% defectors. However, when comparing half of the surfactant production by the inducible strain alone, the results are not significantly different (see Materials and methods for statistical analysis) from the rhamnose levels produced by the inducible strain mixed with 50% defectors, indicating that the surfactant production by the inducible strain is not affected by the presence of non-producing defectors (Supplementary Figure S1).

The mixed-strain competitions show that for every initial mixing proportion the proportion of cooperators decreased after competition, revealing that cooperators are disfavored by individual-level selection (Figure 3B). The change in cooperator proportion as a function of the initial proportion of cooperators exhibits an inverse bell shape that is typical for a trait that is disfavored in competition (Chuang *et al*, 2009). The population fitness, however, increased with the initial proportion of cooperators, confirming that cooperators benefit the entire colony (Figure 3C). Also, notable was that fitness plateaued for high initial cooperator proportions (above 0.5) indicating diminishing returns (Figure 3C).

## Theory for multilevel selection

The swarming competitions using an induced cooperator (Figure 3) reveal that swarming cooperation benefits the population and would thus be favored by group-level selection. However, cooperators are outcompeted within colonies, which means that swarming is disfavored by individual-level selection. In nature, selection can occur simultaneously at multiple levels and the balance between the different levels ultimately determines the evolutionary fate of social traits (Keller, 1999). Could there be situations under which multilevel selection favors costly swarming cooperation?

To answer this question, we consider a theoretical scenario introduced by Hamilton (1975) and more recently applied by

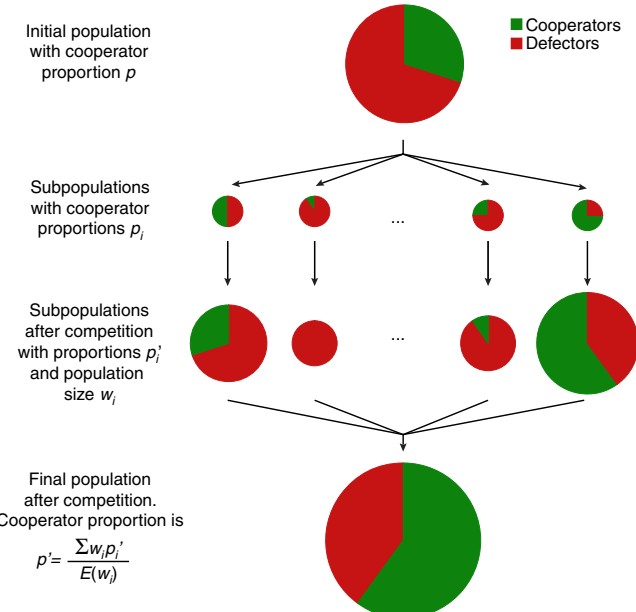

**Figure 4** The theoretical scenario used in multilevel selection analysis. A global population with an initial proportion of cooperators $p$ is sampled to seed several subpopulations each with an initial local cooperator proportion $p_i$ (the average across all populations being $p$). Each subpopulation is allowed to compete for 24 h and at the end of the competition the proportion of cooperators has changed to $p_i'$. In swarming motility, the subpopulations seeded with higher initial proportions of cooperators tend to produce larger colonies ($w_i$ increases with $p_i$), and thus will contribute more to the global pool. After competition, all subpopulations are pooled together and the final proportion of cooperators is assessed, $p' = \sum_i (w_i p_i)/\bar{w}$. Cooperation is favored by multilevel selection when $\Delta p = p - p' > 0$.

Chuang *et al* (2009) to investigate cooperation in a synthetic microbial system. A population of cooperators and defectors (the global population) is distributed heterogeneously into colonies (or local subpopulations) of varying mixing proportions. Each colony is allowed to compete and, at the end, all colonies are pooled together. The change in the total proportion of cooperators in the global population is given by the average across all subpopulations (Figure 4). It is important to note that this average is a weighted average that takes into account the fitness of each individual colony, because colonies with greater fitness contribute more to the global pool.

The approach is based on the multilevel selection framework of the Price equation (Price, 1970). Applied to our system, the Price equation partitions the global change in the proportion of cooperators, $\Delta p$, into its group-level and individual-level selection components:

$$E(w_i)\Delta p = \mathrm{cov}(p_i, w_i) + E(w_i \Delta p_i) \qquad [1]$$

Here, $w_i$ is the fitness of a colony $i$ and its value is a function of the proportion at which cooperators are initially mixed with defectors in that colony ($p_i$). The change in cooperator proportion within the colony, $\Delta p_i$, quantifies individual-level selection. $E(w_i)$ is the average fitness across all colonies that make up the global population. The Price equation highlights that, assuming that the initial population density is the same in all subpopulations, we need only to know the following three quantities to determine whether cooperators are favored by multilevel selection ($\Delta p > 0$):

1. The change in cooperator proportion in a colony as a function of the mixing proportion, $\Delta p_i(p_i)$.

2. The colony fitness as a function of the mixing proportion, $w_i(p_i)$.
3. The distribution of cooperators across all colonies.

Points 1 and 2 are addressed using data from our swarming competitions (Figure 3B and C). To interpolate the data for the entire range $0 \leqslant p_i \leqslant 1$, we developed a population dynamics model (see Materials and methods) that we parameterized by fitting the experimental data (lines in Figure 3B and C). Although simpler statistical regressions could also be used to interpolate such data (Smith *et al*, 2010), a population dynamics model offers the advantage that the parameters obtained provide mechanistic insight. We tested the model by fitting two additional sets of competition experiments carried out at 0% L-arabinose (low levels of cooperation) and 4% L-arabinose (high levels of cooperation; Supplementary Figure S2). The parameters allowed us to calculate relative growth yields of cooperators and defectors as a function of the level of biosurfactant induction (Figure 3D). Consistent with induced surfactant synthesis carrying a metabolic burden (Xavier *et al*, 2011), the growth yield of cooperators decreased relative to that of defectors with the increasing level of surfactant induction. When the mathematical model is suitably parameterized (Supplementary Table 1) it provides functions for $\Delta p_i(p_i)$ and $w_i(p_i)$.

Point 3, the distribution of mixing proportions across all colonies, is crucial because it sets the genetic relatedness between the actors and recipients. Relatedness, as defined in Hamilton's kin selection (Hamilton, 1964), quantifies the similarity of actors and recipients at loci relevant to cooperation (Smith *et al*, 2010). Relatedness is calculated here from the

distribution of cooperator mixing proportions in the subpopulations using the following expression (Grafen, 1985; Damore and Gore, 2012):

$$r = \frac{\mathrm{Var}(p_i)}{E(p_i) - E(p_i)^2} \qquad [2]$$

Equation 2 highlights the importance of the mean, $E(p_i)$, and variance, $\mathrm{Var}(p_i)$, of the mixing proportions seeding the colonies. In the absence of any process generating variance, all colonies will be founded with the same proportion as the global population, $E(p_i) = p$. In this situation, relatedness is null ($r = 0$), individual-level selection prevails and costly cooperation is disfavored. When variance is extremely high and the two strains segregate entirely, for example if a single cell seeds each colony, relatedness is maximal ($r = 1$) and group-level selection prevails. In real systems, relatedness may have intermediate values and be generated by different processes such as population bottlenecks (Chuang *et al*, 2009; Nadell *et al*, 2010), by cooperators physically sticking to each other (Smukalla *et al*, 2008) or other sources of population viscosity (Queller, 1994). Here, we keep our analysis general by using a statistical model based on a log-normal distribution for the ratio of cooperators to defectors (the logarithm of the cooperator-to-defector ratio follows a normal distribution with mean µ and standard deviation σ; see Materials and methods). Defined this way, the probability density function for mixing proportions, named $f(p_i)$, has the convenient feature that it transits gradually from a unimodal distribution (which corresponds to low segregation) to a bimodal distribution (strong segregation) by increasing a single parameter, σ (Figure 5A). The variance is a monotonically increasing function of σ (see Materials and methods) and, consequently, relatedness increases with σ (Equation 2, note that $E(p_i) = p \approx 1/(1 + 10^{-\mu})$). This model of population structure allows simulating scenarios where the mean fraction of cooperators varies while σ remains constant. We would expect σ to remain constant whenever there is a fixed population viscosity at small-length scales that is negligible for within-colony cooperation but high enough to influence seeding of the next round of competitions.

With the three tools in hand, $\Delta p_i(p_i)$, $w_i(p_i)$ and $f(p_i)$, we investigated multilevel selection computationally in the following way: for a given proportion of cooperators in the global population, $p$, we generate subpopulations according to $f(p_i)$. We then use $\Delta p_i(p_i)$ and $w_i(p_i)$ to calculate the competition outcome in each subpopulation. Finally, we calculate the change in the global proportion of cooperators, $\Delta p$, using Price equation. Again, the final cooperator proportion corresponds to a simple calculation of the weighted average of $p_i$ across all subpopulations using the normalized colony fitness, $w_i/E(w_i)$, as the weight parameter (equation 1).

## Multilevel selection requires high relatedness to favor costly swarming

To understand whether multilevel selection can favor swarming cooperation, we first investigated a population with a global proportion of cooperators equal to 0.5 (Figure 5B). Multilevel selection simulations showed, as expected, that cooperators are outcompeted when the variance across all colonies is low (low σ, corresponding to a low relatedness). Keeping the global proportion fixed at 0.5 but increasing the variance increases cooperation advantage progressively: with higher variance, it is more likely to have colonies seeded with large proportions of cooperators, which favors cooperation. Cooperators are eventually favored when σ > 1 ($r > 0.53$).

We tested the simulation results with direct calculations from experimental data. These calculations did not rely on the mathematical model but were rather carried out by re-sampling the swarming competition data (Figure 3B and C) to re-create global populations with cooperator proportions of 0.5 and four different relatedness values (see Materials and methods). The calculations confirmed our simulation results (Figure 5B, data points).

Finally, we simulated multilevel selection for the full range of global cooperator proportions (Figure 5C). The simulations revealed that, because of the diminishing returns noted earlier (Figure 3C), intermediate values of σ produce an evolutionary equilibrium where cooperators and defectors coexist (Supplementary Figure S3B). The diminishing returns make

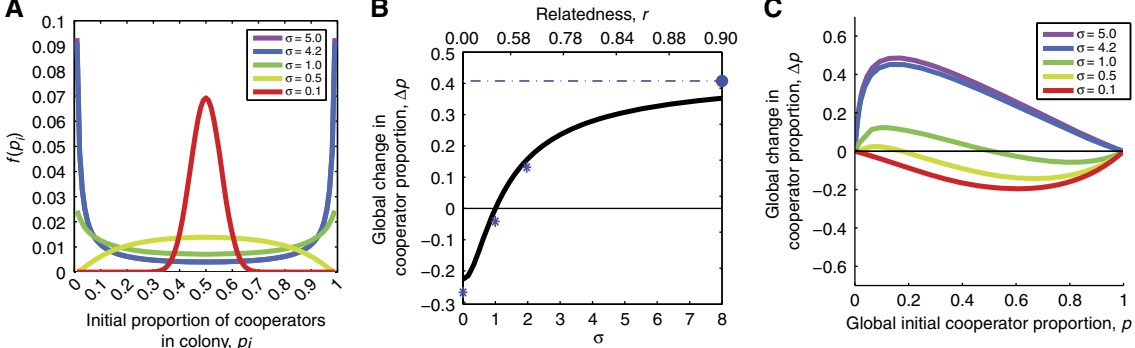

**Figure 5** Multilevel selection favors costly swarming only when relatedness is high. (**A**) Probability density function, $f(p_i)$, used to simulate variance in mixing proportions of cooperators. Parameter σ gradually changes the distribution from unimodal (low relatedness) to bimodal (high relatedness). (**B**) Multilevel selection simulations for a global proportion of cooperators of $P = 0.5$ reveal that σ > 1 favors cooperation. Simulations were compared with direct calculations from re-sampled data (blue stars and dashed line with circle, see Materials and methods). (**C**) Multilevel selection analysis for the entire range of global proportion of cooperators, $0 \leqslant p \leqslant 1$, reveals that cooperation can reach fixation when σ > 4.2.

it particularly difficult for cooperation to spread beyond the fraction at which the diminishing returns set in and, therefore, cooperators only reach fixation ($\Delta p > 0$ for all values of $p$) when $\sigma > 4.2$ (Supplementary Figure S3C). The corresponding value of relatedness is $r > 0.84$ (calculated for $p = 0.5$), which is a very high value. Intuitively, this means that if the process generating relatedness were stochastic sampling of a diluted population (Chuang *et al*, 2009) then subpopulations would have to be inoculated with an average of 0.65 cells/colony, an extremely low number. Swarming colonies inoculated by extremely low numbers of cells could potentially lead to very different results (a subject that we will address in a future study). The requirement for high relatedness is due to a combination of strong individual-level selection against cooperation (Figure 3B) and diminishing returns in the group-level benefits (Figure 3C; Supplementary Figure S3B). Consequently, costly swarming cooperation can be favored by multilevel selection to the point of reaching fixation but only when the genetic relatedness is very high.

## Expansion-driven versus growth-driven cooperation

The calibrated multilevel selection model was also used to test new hypotheses computationally by simulating new competitions beyond the experimental conditions. We asked whether swarming cooperation could still be favored when subpopulations competed for longer periods. Simulations predict that longer competitions would not affect the outcome and that after 48-h competitions cooperators could still reach fixation for $\sigma > 4.2$ (Supplementary Figure S4). We compared these simulations with a different cooperative scenario inspired by the study by Chuang *et al* (2009): growth-driven cooperation. In that system, a costly extracellular signal induces antibiotic resistance in its recipients. Such a trait boosts the population growth rate but does not cause population expansion. Therefore, as Chuang *et al* (2009) noted, the benefits are only felt if the competitions end during the limited time span of exponential growth. Whereas the benefits of swarming cooperation can last beyond the exponential growth phase, the population benefits of growth-driven cooperation should be transient (Figure 6A, see Materials and methods for model implementation).

Our simulations of growth-driven cooperation showed indeed that when growth-driven subpopulations are allowed to reach carrying capacity, such as in 48-h long simulations, the benefits of cooperation are lost and the cooperator strain loses globally irrespective of the relatedness value (Figure 6C). In summary, purely growth-driven cooperation, which does not increase carrying capacity, is likely to be disfavored when competitions are long enough. In contrast, swarming

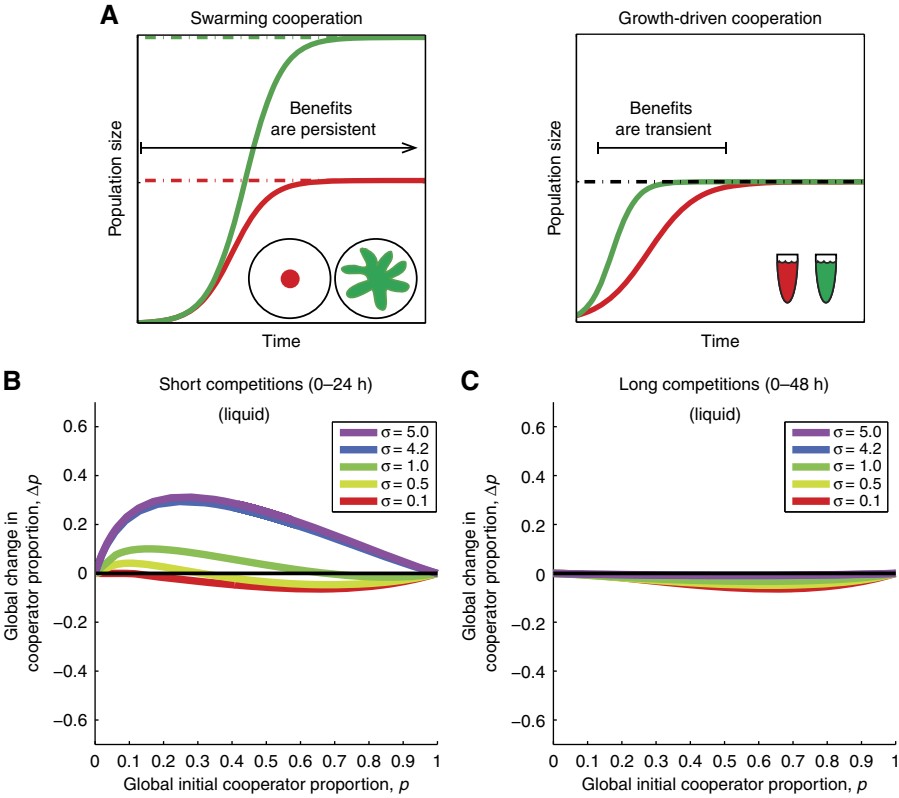

**Figure 6** Growth-driven cooperation has transient benefits and is disfavored by multilevel selection when competitions are long. (**A**) Schematic illustrating that while swarming expands population carrying capacity and its benefits persist even after the growth phase, growth-driven cooperation, as demonstrated by Chuang *et al* (2009) on the other hand, boosts growth rate but its benefits are restricted to the duration of the growth phase. Red line represents defectors and green represents cooperator populations. (**B**) Growth-driven cooperation can be favored after short competitions lasting only during exponential phase. (**C**) If the competitions run long enough such that subpopulations reach carrying capacity, then the benefits of cooperation vanish and cooperation is disfavored.

cooperation is robust to longer competitions because spatial expansion increases population carrying capacity, thus benefiting the colony in a time-independent way.

## Prudent cooperation is widely favored by multilevel selection

Our multilevel selection analysis shows that swarming cooperation by a constitutive cooperator can be favored to fixation even when it is costly, but only when relatedness is high (Figure 5C). In the absence of an active mechanism to increase relatedness such as kin recognition or high population viscosity, the dependence on high relatedness is a stringent constraint that makes costly swarming unlikely in natural populations where strains and species mix (West *et al*, 2006). We therefore investigated the effect of multilevel selection on metabolic prudence, the native regulatory mechanism of rhamnolipid secretion in *P. aeruginosa* (Figure 1A). We carried out experimental competitions at a range of mixing proportions between the wild-type and the

defector strain and, as we had done before for the inducible cooperator, we measured changes in wild-type proportion and colony fitnesses. The examination of a wide range of mixing proportions revealed that, in contrast to induced cooperators (Figure 3B and C), prudent wild-type cooperators were not disfavored by individual-level selection but actually had a marginal advantage (Figure 7A). Moreover, colony fitness increases steadily with the initial proportion of cooperators, showing comparatively little signs of diminishing returns (Figure 7B). We parameterized our dynamic model of swarming competitions for the wild type (Figure 7A and B, lines). In support of previous results (Xavier *et al*, 2011), the growth yields obtained from model fitting showed that, unlike induced cooperation, metabolic prudence can increase colony fitness without a detriment to cooperator yields (Figure 7C). We then analyzed multilevel selection. As expected from the marginal advantage in individual-level selection and the absence of diminishing returns, multilevel selection favored wild-type cooperators at all levels of variance in subpopulation mixing proportions. We also investigated whether the small advantage is essential for the success for wild-type

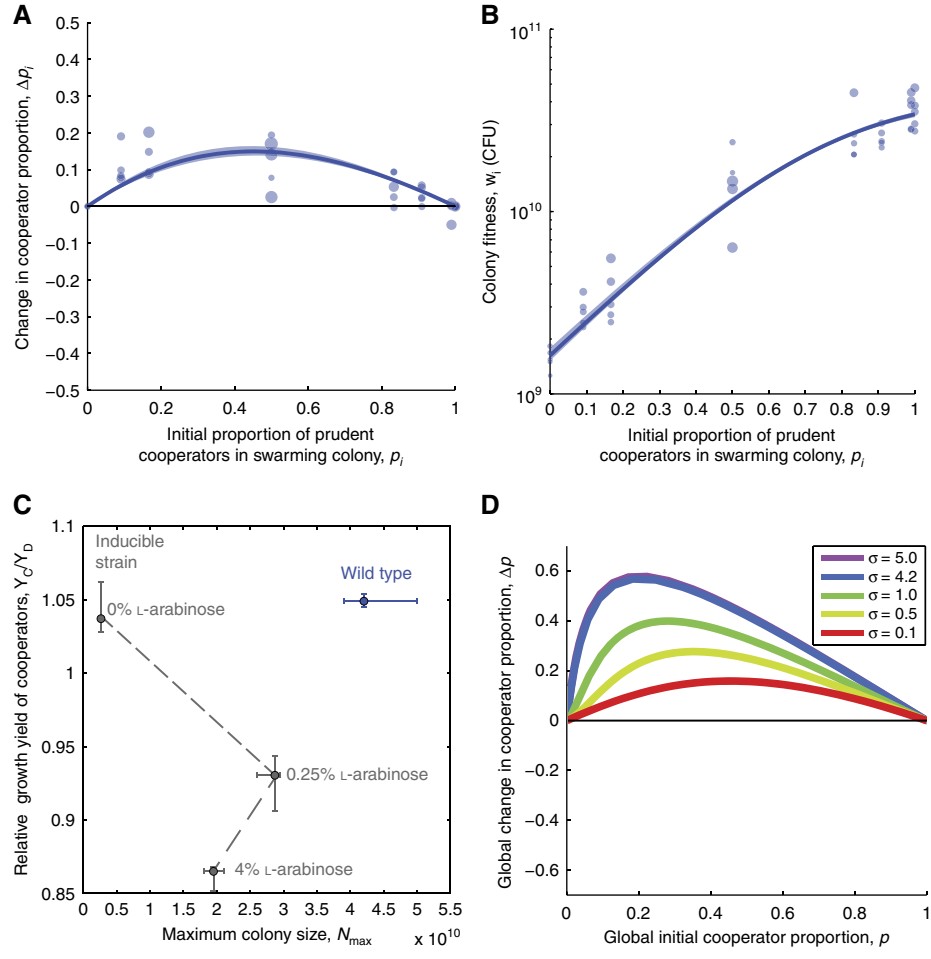

**Figure 7** Metabolic prudence is favored more widely than induced cooperation. (**A**) The change in prudent cooperator (wild-type) proportion due to swarming competitions shows a marginal individual-level advantage of prudent cooperators. (**B**) Colony fitness increases monotonically with cooperator proportion. (**C**) Comparison of calculated yields and maximum colony size between prudent (wild-type) and inducible cooperators induced at three levels of L-arabinose. Only the wild-type strain has both high yield and large colony expansion. (**D**) Multilevel selection of wild-type against the defector strain reveals that prudent cooperators are favored for all values of relatedness.

cooperators. The results show that even if wild-type cells had no advantage at the individual level and had the same yield as defectors, the wild-type strain would still be favored after multi-level selection thanks to the absence of diminishing returns (Supplementary Figure S5). Furthermore, increasing colony variance (higher relatedness) enhances the advantage of prudent cooperators both in the original scenario where wild-type cells have an individual-level advantage and in the neutral theoretical case (Figure 7D and Supplementary Figure S5, respectively).

## Conclusion

The evolution of cooperative traits is a central problem in biology (Pennisi, 2005). What are the evolutionary mechanisms favoring cooperation when costly cooperative traits that benefit other individuals cannot be favored by selection acting at the individual level alone? Microbial systems are becoming increasingly popular as models to address this question experimentally and in a quantitative way thanks to large population sizes, short generation times and the ability to manipulate traits genetically (West *et al*, 2006; Xavier, 2011b; Damore and Gore, 2012; Celiker and Gore, 2012a).

But while social evolution is learning considerably from the interface with microbiology the reverse may not be true. In the microbiology literature, social traits are often explained using group-selection arguments (Costerton *et al*, 1999; Monds and O'Toole, 2009; Lee *et al*, 2010). This logic, sometimes referred in the evolutionary literature as the original or 'old' type of group selection (West *et al*, 2007), was introduced by Wynne-Edwards in the 1960s and argues that cooperation is favored because groups of cooperative individuals are fitter than groups of selfish individuals (Wynne-Edwards, 1963). However, the same logic neglects that traits that are costly to individuals would be outcompeted within group selection (West *et al*, 2006; Foster and Bell, 2012). The solution to this problem comes from recognizing that natural selection acts at multiple levels of biological organization, an idea first introduced in Hamilton's kin selection (Hamilton, 1964), and 'modern' forms of group selection (Traulsen and Nowak, 2006) are equivalent (Lehmann *et al*, 2007). The Price equation (Equation 1) is one generalization of this idea that, as we illustrate here, is particularly suited for the quantitative analysis of microbial social traits. Recognizing that selection acts at multiple levels is key to analyzing microbial social traits (West *et al*, 2006) can help reveal novel mechanisms stabilizing cooperation (e.g., Foster *et al*, 2004; Xavier and Foster, 2007; Ostrowski *et al*, 2008; Nadell *et al*, 2010; Nadell and Bassler, 2011; Xavier *et al*, 2011; Dandekar *et al*, 2012; Celiker and Gore, 2012b; Koschwanez *et al*, 2013), and can eventually inspire therapeutic strategies against pathogens (Foster, 2005; Brown *et al*, 2009; Boyle *et al*, 2013).

We combined quantitative experiments with mathematical modeling to identify conditions favoring costly cooperative swarming *P. aeruginosa* (Figure 3B and C). Using an engineered strain enabled us to manipulate cooperation and to show that in spite of individual-level costs, costly swarming can still be favored under specific conditions. However, the conditions allowing the spread of constitutive cooperation are

limited. This is in part due to high costs of cooperation but also, to large extent, due to strong diminishing returns (Figure 3C), a feature that is likely to be common in cooperative systems (Foster, 2004). Because population-level benefits of constitutive cooperation level off when the cooperator proportion is $p > 0.5$, it is difficult for cooperation to spread beyond this value (Supplementary Figure S3B). As a consequence, the evolution of constitutive cooperation can require extremely high relatedness, which is unlikely in natural populations where strains and species mix. We also show that metabolic prudence, the native mechanism to regulate rhamnolipid synthesis in *P. aeruginosa* (Xavier *et al*, 2011), is particularly advantageous because it provides a small direct benefit to cooperators (Figure 7A) and lacks the strong diminishing returns (Figure 7B). The marginal benefit of metabolic prudence in within-colony competitions is amplified by multilevel selection (Figure 7D), and greatly expands the conditions favoring swarming cooperation.

In our multilevel selection simulations, we assumed that the subdivision of the global population into subpopulations follows a distribution with the same variance at every round. This is a common assumption in social evolution models that can be plausible under certain conditions, for example, limited dispersal. It is also possible, however, that the variance in the formation of subpopulations would change from one competition round to the next, causing the distribution of subpopulations to fluctuate in time. In this scenario, we expect that the evolution of the wild-type strain would be favored even more compared with the inducible cooperator strain, because the wild type is able to win in each competition against defectors irrespective of how the global population is sampled.

A notable conclusion from our study is that the spatial expansion caused by swarming provides persistent benefits to a population (Figure 6C; Supplementary Figure S4). Population growth is always limited by a carrying capacity (e.g., Brown *et al*, 2004). Cooperative traits that, unlike swarming, boost growth rate without expanding carrying capacity can only have transient benefits, and therefore are less likely to be favored by multilevel selection when competitions are long (Figure 6A). In fact, Chuang *et al* (2009) had noted that, in their synthetic system, cooperators were favored by multilevel selection only if the subpopulations were sampled during the exponential growth phase; once the populations reached stationary phase the cooperative benefits would disappear. These results also relate to the long-studied differences between $r$ and $K$ selection, where $r$ refers to the maximal intrinsic growth rate and $K$ refers to carrying capacity (Pianka, 1970; MacArthur and Wilson, 2001). An $r$-strategy depends on traits associated with rapid growth (high fecundity, early maturity and short generation times) and is suitable for primary colonizers of new environments (Loya, 1976). However, in stable environments, quality overcomes quantity and $K$-strategies that grow slower but have larger carrying capacity have the competitive advantage and can displace $r$-selected species (Pianka, 1970). In the case of social traits, a cooperative trait that only increases growth rate is comparable to an $r$-strategy as its benefits are transient. Meanwhile, a cooperative trait that benefits the population by allowing it to grow to a larger carrying capacity is comparable to a

*K*-strategy. Our work is also consistent with theory (Lehmann *et al*, 2006; Alizon *et al*, 2008; Houchmandzadeh and Vallade, 2012) and recent experimental studies (Datta *et al*, 2013; Van Dyken *et al*, 2013) showing that population range expansion can be a key factor in stabilizing the evolution of cooperation.

There are many examples of microbial social traits that expand carrying capacity even without explicit spatial expansion. Sharing of iron scavenging molecules (West *et al*, 2006) and digestive enzymes (Greig and Travisano, 2004; Griffin *et al*, 2004; Gore *et al*, 2009) can expand the achievable size of a nutrient limited population in shaken test tubes. Similarly, high-yield metabolic pathways allow a population to make more efficient use of nutrients and thus to achieve higher numbers in spite of slower growth (Pfeiffer *et al*, 2001; Kreft, 2004; Frank, 2010). Expansion of carrying capacity can thus be achieved by many different means in addition to spatial spreading. We therefore expect that cooperation by prudent expansion of carrying capacity, rather than fast growth, is more stable and more commonly found in nature.

# Materials and methods

## Bacterial strains and swarming assays

The construction of the *Pseudomonas aeruginosa* strains and their GFP and DsRed-Express varieties was described previously (Xavier *et al*, 2011). All bacteria were cultured in LB (Lysogeny Broth) liquid medium overnight, followed by triple washing 1 ml with saline buffer. Labeled strains were mixed at different ratios to inoculate in Petri dishes with soft agar (0.5% agar). Minimum medium for soft agar plates was prepared as described previously (Xavier *et al*, 2011) with the addition of L-arabinose when needed. Serial dilutions were done for each of the cell mixes and CFUs (colony forming units) were also counted out as previously described (Xavier *et al*, 2011). All incubations were at 37°C. Fluorescently labeled strains were counted from plates imaged with fluorescent scanner Amershan Typhoon 9400 (GE Healthcare). Color pictures of swarming colonies were obtained using the same fluorescent scanner.

## Mathematical modeling

We model swarming competitions as two strains, cooperators (*C*) and defectors (*D*), competing for a finite nutrient source (*N*) in a closed system representing the Petri dish. The model consists of three simple ordinary differential equations:

$$\frac{dC}{dt} = qY_C \frac{N}{N+K_n}C$$
$$\frac{dD}{dt} = qY_D \frac{N}{N+K_n}D \qquad [3]$$
$$\frac{dN}{dt} = -q\frac{N}{N+K_n}(C+D)$$

which implement Monod growth kinetics with $K_n$ as the half-saturation constant (Monod, 1949). Both competing strains consume nutrients at the same rate, $q$, but cooperators may have a lower growth yield than defectors, $Y_C < Y_D$, if biosurfactant production is costly. If we now consider that $K_n \ll N_0$ for $N \neq 0$:

$$\frac{dC}{dt} = qY_C C$$
$$\frac{dD}{dt} = qY_D D \qquad [4]$$
$$\frac{dN}{dt} = -q(C+D)$$

if $N = 0$, then growth halts and $\frac{dC}{dt} = 0$, $\frac{dD}{dt} = 0$, $\frac{dN}{dt} = 0$. Therefore, both strains grow exponentially while nutrients are available but stop growing once nutrients run out (zeroth order kinetics). We assume that the maximum amount of nutrients a colony will have access to ($N_0$) is determined at the very beginning of the competition by the proportion of cooperators in the initial mixing of the two competing strains,

$p_i = C_0/(C_0 + D_0)$. We observed empirically that when $p_i \leqslant 0.5$, the achieved population size, and therefore $N_0$, increases quickly with small $p_i$ increments, however saturates for $p_i \geqslant 0.5$ (Figure 3b). Therefore, we describe $N_0$ as a sigmoidal function that captures our empirical observation of diminishing returns:

$$N_0(p_i) = \frac{N_{\max}N_{\min}e^{gp_i}}{N_{\max} + N_{\min}(e^{gp_i} - 1)} \qquad [5]$$

$N_{\min}$ represents the amount of nutrients that a non-swarming population ($p_i = 0$) can take up. $N_{\max}$ represents the maximum amount of nutrients that a swarming colony composed entirely of cooperators ($p_i = 1$) can consume. The coefficient $g$ is the rate by which available nutrients increase as a function of $p_i$. To simulate growth-driven cooperation (Figure 4), we used a modified version of our model such that the growth rates, not the available nutrients, increase in the presence of cooperators. $N_0$ is constant and the rate of nutrient uptake $q$ is a function of $p_i$:

$$q(p_i) = q_D + (q_C - q_D)p_i \qquad [6]$$

$q_D$ and $q_C$ are defined as the rate at which a population of only defectors or only cooperators takes up nutrients, respectively. In the simulations of growth-driven cooperation, we set $q_D = 0.5$ and $q_C = 1$ as a population of pure cooperators has a higher growth rate than a population of pure defectors. All simulations and parameter fitting (see supporting Table I) were carried out in Matlab (R2011a, the Mathworks).

## Probability density function

The probability density function $f(p_i)$ is defined such that that the ratio of cooperators to defectors, $C/D$, follows a log-normal distribution. The random variable $X = \log_{10}(C/D)$ is therefore normally distributed ($X \sim N(\mu, \sigma)$); and random variable $p_i = C/(C + D)$, the proportion of cooperators in the population, is a function of $X$ such that $p_i = g(X) = (1 + 10^{-X})^{-1}$. The variance and mean of $p_i$ were approximated through the Delta method using first-order Taylor expansions of $g(x)$ around $X = \mu$. $E(p_i) \approx (1 + 10^{-\mu})^{-1}$ and $Var(p_i) \approx g'(\mu)^2\sigma^2$. The probability density function of $p_i$ has the analytical expression:

$$f(p_i) = \frac{1}{\sigma\sqrt{2\pi}}\exp\left(-\frac{1}{2}\left(\frac{\log_{10}(p_i/(1-p_i)) - \mu}{\sigma}\right)^2\right) \times \left|\frac{1}{\ln(10)(p_i - p_i^2)}\right| \qquad [7]$$

## Direct calculations of multilevel selection

To confirm the prediction from multilevel selection (Figure 5B), we re-sampled experimental data by selecting equal number of data points with initial cooperator proportion ($p_i$) of 17 and 83% to generate a global population with initial $p = 0.5$. The final proportion of cooperators was used to calculate the global $\Delta p$ as well and the relatedness value corresponding to $\sigma$. The same was done using the data points from initial local cooperator proportions ($p_i$) of 1 and 99%. The data corresponding to initial $p_i = 0$ and $p_i = 1$ were used to determine the asymptotic value of $\Delta p$ for $r = 1$.

## Surfactant quantification

The relative amount of rhamnolipids produced by the inducible strain in the absence and presence of defectors was quantified using an anthrone assay (Xavier *et al*, 2011) to assess rhamnose production in the defector, the wild-type and the inducible strains alone, as well as in a mix between the inducible strain + 50% defectors in a liquid medium with the same composition of the swarming media. The rhamnose levels were normalized by subtracting the average amount detected from the defector strain (background) and dividing by the average amount produced by the wild-type strain. We used a Wilcoxon rank-sum test to evaluate whether rhamnolipid production by the inducible

strain alone was significantly different from production in a mix with 50% defectors. The significance test showed that the inducible strain's rhamnolipid production is different from the inducible + 50% defector strain with $P$-value = 0.028. We also found that when comparing half of the amount of secreted rhamnose by the inducible strain with that of the mix with 50% defectors, there was no significant difference with a $P$-value of 0.382.

## Supplementary information

## Acknowledgements

We thank Kevin Foster, Giles Hooker, Edo Kussel, Silja Heilman, Dave van Ditmarsch, Carey Nadell, Carlos Carmona Fontaine, Vanni Bucci and Will Chang for comments and helpful discussions. This work was supported by a seed grant from the Lucille Castori Center for Microbes, Inflammation, and Cancer and by the Office of the Director, National Institutes of Health of the National Institutes of Health under Award Number DP2OD008440 to JBX. The content is solely the responsibility of the authors and does not necessarily represent the official views of the National Institutes of Health.

*Author contributions:* LVR and KEB performed experiments. LVR and JBX designed the study, analyzed the data and created the mathematical model. LVR performed computer simulations. LVR, KEB and JBX wrote manuscript and prepared figures.

## Conflict of interest

The authors declare that they have no conflict of interest.

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
