## [Review Process File · Molecular Systems Biology]

Multilevel selection analysis of a microbial social trait

Laura De Vargas Roditi, Kerry E. Boyle, Joao B. Xavier

Corresponding author: Joao B. Xavier, Memorial Sloan-Kettering Cancer Center

Review timeline:

Submission date:	17 May 2013
Editorial Decision:	24 June 2013
Revision received:	12 July 2013
Editorial Decision:	18 July 2013
Revision received:	19 July 2013
Accepted:	24 July 2013

Editor: Maria Polychronidou

Transaction Report:

1st Editorial Decision

24 June 2013

Thank you again for submitting your work to Molecular Systems Biology. We have now heard back from the three referees who accepted to evaluate the study. As you will see, the referees find the topic of your study of potential interest and are supportive. However, they raise a series of concerns and make suggestions for modifications, which we would ask you to carefully address in a revision of the present work. Reviewer #3 suggests elaborating on the comparison of engineered vs. wild type strains and growth- vs. expansion-driven cooperation, which would certainly add to the completeness of the study.

Please resubmit your revised manuscript online, with a covering letter listing amendments and responses to each point raised by the referees. Please resubmit the paper ****within one month**** and ideally as soon as possible. If we do not receive the revised manuscript within this time period, the file might be closed and any subsequent resubmission would be treated as a new manuscript. Please use the Manuscript Number (above) in all correspondence.

REFeree REPORTS:

Reviewer #1 :

I have reviewed this manuscript for another journal and have found that the authors worked hard to address my questions and concerns. I have tried to look for changes that are relevant for my review, but apologies to the authors if they have made changes that I have missed.

The authors present a nice combination of experiments, theory, and simulations to explore the

conditions under which multilevel selection will favor cooperation. They show that spatial population expansion can favor cooperation by group selection, but only when genetic relatedness is sufficiently high. This basic result is perhaps not particularly surprising, but the authors have done a good job of analyzing their system and clarifying the issues involved. In addition, I appreciated the simple yet mechanistic modeling approach that was used. This combination of experiment and modeling will be of interest to researchers in the field.

The primary complaint that one might have is that the primary conclusion is one that in some sense has to be true. Assuming that swarming motility is prone to cheating behaviors based on individual level selection, and that the presence of such cheaters limits the overall population size, it will always be the case that sufficiently strong group-level selection will favor cooperation. The interesting results are how this occurs, and in particular the interplay between assortment, group fitness, and the final outcome.

I believe that the diminishing returns that the authors observe experimentally will likely be a common feature of many cooperative systems. The authors could comment that it is difficult for cooperation to spread beyond the fraction at which the diminishing returns sets in.

I find analyses based on relatedness and Price's equation unnecessarily complicated and opaque, and in addition are difficult to use correctly. Given that the primary contribution of this study is the computational analysis, I would like the authors to point out very clearly that the change in the total fraction of cooperation can be obtained from a direct average over all of the sub-colonies. This is more straight-forward to implement and avoids complicated terminology and definitions. I strongly believe that in general researchers should use this direct approach, as I feel that it is more reliable and provides more insight (but of course the authors can disagree!).

I think that it would also be interesting if the authors explored what happens for different initial seeding densities, particularly since bottlenecks are a common way to generate the variation in cooperator fraction that group selection acts upon.

Other points:

1. line 105: "where then imaged" -> "were then imaged"
2. line 205: I feel that the starting "Furthermore" is not really appropriate. The sentence is a disclaimer rather than a statement further supporting the previous sentence, right?
3. Over the last couple of months there have been two papers exploring the effect of range expansions on cooperation in microbial populations: Michael Desai in *Current Biology* and Jeff Gore in *PNAS*. These should be discussed.

Reviewer #2 :

This paper examines the conditions required for cooperative swarming to remain evolutionary stable. The authors combine experimental work with mathematical modeling, using a multi-level selection approach. This is a nice and concise paper, which greatly adds to our understanding of the forces that are favorable or disruptive for cooperative swarming, and microbial cooperative traits in general. Previous work (by some of the authors) has shown that biosurfactant production is delayed to the onset of the stationary phase. This delay seems adaptive because the relative cost of biosurfactant production is reduced in this growth phase, which prevents the invasion of cheating non-biosurfactant producers. In the current paper, the authors chose an elegant approach by engineering a strain, which produces biosurfactants constitutively throughout the growth cycle. This was done to demonstrate more generally how individual-level selection can disrupt group-level benefits. I must admit that I have reviewed this paper before for another journal. I am pleased to see that the authors have implemented some of my previous concerns. I therefore have only a few minor points I wish to see addressed.

- 1) Some evolutionary terms are used in a confusing way.
L254. "How can a trait that benefits other individual be favored in the face of natural selection for

selfishness?" This question assumes that natural selection always favors selfish behavior. This is incorrect. Natural selection favors individuals that maximize their inclusive fitness (i.e. the sum of benefits to self and kin). In other words, natural selection is the overarching force behind evolution, and can be split into different levels (individual-level, group-level, etc. selective forces). Please revise.

Related to this:

L15. "...swarming cooperation is favorable for multilevel selection..." It is the other way round. Multi-level selection can explain why swarming cooperation is favored. But, in the context of the above sentence the authors probably want to say "group-level selection could favor cooperative swarming because it causes population expansion."

L16. "However, this strong group-level benefit does not invariably result in natural selection." Again, this reads odd. It is not group-level benefits that drive natural selection. Group-level benefits can drive the evolution of cooperative swarming. Again, in the context of the above sentence the authors probably want to say "However, this strong group-level benefit does not invariably result in cooperative swarming being favored by natural selection."

2) As already pointed out in my previous review, I wish to see a more balanced discussion of how the authors' work relates to recent studies that have chosen similar approaches with other microbial cooperative traits. For example, key systems include fruiting body formation in *Dictyostelium discoideum* (reviewed in Strassmann & Queller 2011 PNAS) and *Myxococcus xanthus* (Smith et al. 2010 Science), as well as quorum sensing (Diggle et al. 2007 Nature), and siderophore secretion in *P. aeruginosa* (Griffin et al. 2004 Nature). Siderophore production is probably the best-studied microbial cooperative trait with studies having examined frequency and density effects (Ross-Gillespie et al. 2007 Am Nat, 2009 Evolution), the role of relatedness (Kummerli et al. 2009 Proc B), population expansion/ local competition (Kummerli et al. 2009 Evolution), and cost/benefit ratios (Kummerli et al. 2009 J. Evol. Biol). Referring to this body of work is essential to send a strong signal to microbiologists that group-level benefits are not sufficient for cooperative traits to be selected for.

3) L57-58. The question "Could there be other processes favoring costly swarming cooperation?" is quite vague. The authors could be more explicit by saying that their previous work focused on cost-reducing factors (i.e. prudent resource use), whereas here they focus on the role of relatedness, whereas costs are kept constant. A reference to Hamilton's rule would make this point even clearer.

4) I found the result of an inverse bell curve in Fig. 3b quite interesting. I had expected to see a negative linear relationship. Why do cheats not perform best at lowest frequency? A more detailed discussion would be helpful.

5) L215: remove "that"

6) L269. In the context of this sentence, the reader might conclude that reference (3) is an example of wrong interpretation. This is of course not the case. Please reconsider phrasing. Instead, reference (8) is a clear case of an erroneous interpretation. This could be discussed in more detail, but is maybe not helpful here.

Reviewer #3 :

SUMMARY OF THE MS

The authors address the question of the relative importance of group-level and individual-level selection in the maintenance of cooperation. They construct a new *P. aeruginosa* strain in which expression levels of biosurfactant are tuned via external concentrations of L-arabinose. In this new system they can control the cost to benefit ratio of the production of biosurfactant. They then study how the "individually costly" but "group beneficial" production of biosurfactant can be selected against invasion by non-producer individuals. To understand the outcome of two-strain competition experiments they propose a Price equation-based model. This enables them to separate group-level and individual-level selection acting on the competition outcome. They conclude that inducible

producer is out-competed by non-producer under high mixing/low relatedness regime, and that high strain segregation is needed for fixation of inducible producer strain. Finally, they compare the tunable producer strain with a wild-type strain that doesn't get out-competed by the non-producer strain even under high mixing regime.

GENERAL COMMENTS

- The main message does not seem to be novel.
- Some results are novel but are not highlighted - specifically the last result on wild type phenotype out-competing non-producer strain.
- Introduction and conclusion give a biased vision of the field.
- Regarding technical aspects, the manuscript is convincing and solid. The engineered inducible producer strain is an useful tool and the authors provide an interesting comparison with the wild-type strain. Only one control is missing: at a fixed arabinose concentration, do engineered cells produce surfactants at a constant rate regardless of the cell density and the presence of non-producers, as expected?

MAJOR CONCERNS:

A major concern is that the main message reported by the authors is not very new and that the introductory background justifying its novelty is partially biased.

1. The introduction and conclusion provide a biased perspective of the literature in the field. 1) Some microbial systems such as *Dictyostelium* in which extensive mixing occurs have evolved a (very) costly cooperation strategy (Fortunato et al., 2003; Sathe et al., 2010). 2) More importantly, cooperation is not explained mainly through group selection in the field, it is rather the opposite; most of the studies propose individual-level selection and kin selection to explain cooperation in microbes (West, Griffin, Gardner, & Diggle, 2006). The background on which the authors base their study is thus not reflecting faithfully the current view on these questions.

2. The novelty of the conclusions is not very obvious. The fact that a constant producer is outcompeted by a non-producer is expected based on previous theoretical studies and published experimental results on bacterial populations (Dao, Kessin, & Ennis, 2000; Doebeli & Hauert, 2005; Rainey & Rainey, 2003; Travisano & Velicer, 2004). The fact that high relatedness favors fixation of cooperators in these situations was also repeatedly shown (Griffin, West, & Buckling, 2004; Nadell, Foster, & Xavier, 2010; West et al., 2006).

3. However the data of Figure 7, illustrating the side-by-side comparison of the engineered and wild type strains, convey a more interesting message than the one highlighted by the authors. They find that a constant production, even at an optimal rate (best group-level fitness vs cost of the cooperator), is way outperformed by the wild-type production scheme. This was not directly predictable based on the previous paper describing the wild-type strain (Xavier, Kim, & Foster, 2011).

4. Results on growth driven cooperation and capacity driven cooperation are interesting but not sufficiently elaborated and are not at all compared to other literature in the area.

SUGGESTED REVISIONS

1. The parts of the introduction and conclusion on individual and group selection in microbes should be revised. This subject has been discussed in many previous studies and the view advocated by the authors is, in our opinion, not under-represented - if the authors think that it is the case, they should provide references.

2. It remains true that the maintenance of cooperation in natural systems is not fully understood. Experiments and theory indicate special conditions (structured environment, low mixing, high relatedness, differential adhesion) under which cooperators can outcompete defectors. On the other

hand, in highly mixed natural populations these conditions are difficultly reached (because mixing is incompatible with high relatedness and structures environment). In this respect, the presence of a mechanism in wt strains for resisting the invasion of defectors (Figure 7) is interesting and would deserve more attention: is this mechanism not an essential factor for maintaining cooperation in this system?

3. Growth vs capacity driven cooperation. This point would also deserve a more elaborate discussion and references to previous studies. Is it for instance related to the distinction between R and K-selection? Can we connect it to the model of <http://www.biomedcentral.com/1471-2148/12/61/>

In our opinion, the paper would benefit from bringing upfront these more original points, rather insisting on the predictable outcomes of competition between inducible producer and defector.

MINOR CONCERNS

1. Figures are mis-numbered (There are 2 figures 3). Some data in the figures are not mentioned nor discussed in the text (Fig 3D). Please either include them in the text or leave them out of the manuscript.

2. Some typos, including in the abstract.

3. It would be very interesting to run simulations while changing the variance (partitioning) at successive rounds. How do various strains displaying different production regulations perform under such selection regimes? Such data might also provide outcomes that would not be easily predictable based on previous studies.

4. The manuscript contains many anthropomorphic expressions that refer to *P. aeruginosa* cells, and in general microbe populations. Although these anthropomorphic expressions are common in many publications in the field, it may be best to avoid them to prevent any misleading interpretation based on their common-sense understanding. These terms include here "altruistic death", "prudent cooperation", "bacterial charity", "defector", "selfish individuals",...

REFERENCES

- Dao, D. N., Kessin, R. H., & Ennis, H. L. (2000, July). Developmental cheating and the evolutionary biology of *Dictyostelium* and *Myxococcus*. *Microbiology* (Reading, England). Retrieved from <http://www.ncbi.nlm.nih.gov/pubmed/10878115>
- Doebeli, M., & Hauert, C. (2005). Models of cooperation Models of cooperation based on the Prisoner's Dilemma and the Snowdrift game. *Ecology letters*, 8, 748-766. doi:10.1111/j.1461-0248.2005.00773.x
- Fortunato, A., Strassmann, J.E., Santorelli, L., Queller, D.C. (2003) Co-occurrence in nature of different clones of the social amoeba, *Dictyostelium discoideum*. *Mol Ecol* 12: 1031-1038.
- Griffin, A. S., West, S. A., & Buckling, A. (2004). Cooperation and competition in pathogenic bacteria. *Nature*, 430(August). doi:10.1038/nature02802.1.
- Nadell, C. D., Foster, K. R., & Xavier, J. B. (2010). Emergence of spatial structure in cell groups and the evolution of cooperation. *PLoS computational biology*, 6(3), e1000716. doi:10.1371/journal.pcbi.1000716
- Rainey, P. B., & Rainey, K. (2003). Evolution of cooperation and conflict in experimental bacterial populations. *Nature*, 425(September), 72-74. doi:10.1038/nature01942.1.
- Sathe, S., Kaushik, S., Lalremruata, A., Aggarwal, R.K., Cavender, J.C., et al. (2010) Genetic heterogeneity in wild isolates of cellular slime mold social groups. *Microb Ecol* 60: 137-148. doi:10.1007/s00248-010-9635-4.
- Travisano, M., & Velicer, G. J. (2004). Strategies of microbial cheater control. *Trends in microbiology*, 12(2), 72-8. doi:10.1016/j.tim.2003.12.009

West, S. a, Griffin, A. S., Gardner, A., & Diggle, S. P. (2006). Social evolution theory for microorganisms. *Nature reviews. Microbiology*, 4(8), 597-607. doi:10.1038/nrmicro1461

Xavier, J. B., Kim, W., & Foster, K. R. (2011). A molecular mechanism that stabilizes cooperative secretions in *Pseudomonas aeruginosa*. *Molecular microbiology*, 79(1), 166-79. doi:10.1111/j.1365-2958.2010.07436.x

1st Revision - authors' response

12 July 2013

Thank you for the review of “Multilevel selection analysis of a microbial social trait”. We are happy to see that the three reviewers agree on the interest of our work and the topic of microbial social evolution. The comments raised are constructive and addressable. We are hereby submitting our revised version that we believe is significantly improved and we hope is suitable for publication in *Molecular Systems Biology*.

Reviewer #1 and #2 report that they had reviewed a previous submission to another journal, and that they are happy that we addressed their earlier comments. Both reviewers raise additional points that we address in this revised version

Reviewer #3 makes several additional comments, which we also address in this revised version. Perhaps most noteworthy is that we carried out the additional control experiments requested by the reviewer to compare surfactant production between the wild-type and the engineered strain and to confirm that surfactants produced by the engineered cells are not affected by the presence of non-producers (new fig. S1). We also carried out additional simulations to demonstrate that the marginal individual-level advantage of the wild-type is important but not essential for its success in multilevel selection. And we also elaborate on the growth- vs. expansion-driven cooperation with appropriate citations as requested, including two recent very papers from the Desai and Gore groups.

We thank all the reviewers for their constructive criticism. Below is a point-by-point reply to the comments.

Reviewer #1 :

I have reviewed this manuscript for another journal and have found that the authors worked hard to address my questions and concerns. I have tried to look for changes that are relevant for my review, but apologies to the authors if they have made changes that I have missed.

The authors present a nice combination of experiments, theory, and simulations to explore the conditions under which multilevel selection will favor cooperation. They show that spatial population expansion can favor cooperation by group selection, but only when genetic relatedness is sufficiently high. This basic result is perhaps not particularly surprising, but the authors have done a good job of analyzing their system and clarifying the issues involved. In addition, I appreciated the simple yet mechanistic modeling approach that was used. This combination of experiment and modeling will be of interest to researchers in the field.

The primary complaint that one might have is that the primary conclusion is one that in some sense has to be true. Assuming that swarming motility is prone to cheating behaviors based on individual level selection, and that the presence of such cheaters limits the overall population size, it will always be the case that sufficiently strong group-level selection will favor cooperation. The interesting results are how this occurs, and in particular the interplay between assortment, group fitness, and the final outcome.

Answer: We are happy to see that the reviewer agrees that our results are interesting and that the combined experimental/modeling work will be of interest to researchers in the field.

I believe that the diminishing returns that the authors observe experimentally will likely be a common feature of many cooperative systems. The authors could comment that it is difficult for cooperation to spread beyond the fraction at which the diminishing returns sets in.

Answer: We agree that diminishing returns are likely to be common in cooperation and that this is well worth highlighting. We improved the text to highlight the importance of diminishing returns. We make this point in the abstract (lines 15 and 18), in the results section (lines 224-227, 274-286) and in the conclusion (lines 308-318) where we now discuss that it can be a common feature.

I find analyses based on relatedness and Price's equation unnecessarily complicated and opaque, and in addition are difficult to use correctly. Given that the primary contribution of this study is the computational analysis, I would like the authors to point out very clearly that the change in the total fraction of cooperation can be obtained from a direct average over all of the sub-colonies. This is more straight-forward to implement and avoids complicated terminology and definitions. I strongly believe that in general researchers should use this direct approach, as I feel that it is more reliable and provides more insight (but of course the authors can disagree!).

Answer: We restructured the section “Theory for multi-level” to address the reviewer’s request for clarity. We now explain first of all and in a clear way that the change in global cooperator fraction is obtained from a *weighed* average (note that this cannot be a simple average, because it needs to take into account the productivity of each individual colony and not only the final fraction of cooperators). Only after that do we proceed with the explanation of the Price equation (we believe this explanation will be valued by the readership of Molecular Systems Biology thanks to the general applicability of the Price equation). That section ends by highlighting once again that the result is equivalent to a weighed average.

I think that it would also be interesting if the authors explored what happens for different initial seeding densities, particularly since bottlenecks are a common way to generate the variation in cooperator fraction that group selection acts upon.

Answer: We thank for the reviewer’s suggestion. We have seen that initial seeding densities do have an effect in the outcome. We do highlight that the analysis carried out here assumes the seeding density is constant (line 157). The effect is interesting but out of the scope of this paper and therefore is the topic of another upcoming paper (under review at the New Journal of Physics, we are happy to supply to the manuscript if needed) – see line 229.

Other points:

1. line 105: "where then imaged" -> "were then imaged"

Answer: The typo was corrected.

2. line 205: *I feel that the starting "Furthermore" is not really appropriate. The sentence is a disclaimer rather than a statement further supporting the previous sentence, right?*

Answer: We deleted “furthermore”. We also explain the subject will be addressed in a future study.

3. *Over the last couple of months there have been two papers exploring the effect of range expansions on cooperation in microbial populations: Michael Desai in Current Biology and Jeff Gore in PNAS. These should be discussed.*

Answer: Thanks for pointing out these recent papers, which are indeed relevant here. We now cite these papers in the discussion (line 345).

Reviewer #2 :

This paper examines the conditions required for cooperative swarming to remain evolutionary stable. The authors combine experimental work with mathematical modeling, using a multi-level selection approach. This is a nice and concise paper, which greatly adds to our understanding of the forces that are favorable or disruptive for cooperative swarming, and microbial cooperative traits in general. Previous work (by some of the authors) has shown that biosurfactant production is delayed to the onset of the stationary phase. This delay seems adaptive because the relative cost of biosurfactant production is reduced in this growth phase, which prevents the invasion of cheating

non-biosurfactant producers. In the current paper, the authors chose an elegant approach by engineering a strain, which produces biosurfactants constitutively throughout the growth cycle. This was done to demonstrate more generally how individual-level selection can disrupt group-level benefits. I must admit that I have reviewed this paper before for another journal. I am pleased to see that the authors have implemented some of my previous concerns. I therefore have only a few minor points I wish to see addressed.

Answer: We are happy that the reviewer agrees on the quality and suitability of our paper, and that it adds to the present understanding of the evolution of cooperation.

1) Some evolutionary terms are used in a confusing way.

L254. "How can a trait that benefits other individual be favored in the face of natural selection for selfishness?" This question assumes that natural selection always favors selfish behavior. This is incorrect. Natural selection favors individuals that maximize their inclusive fitness (i.e. the sum of benefits to self and kin). In other words, natural selection is the overarching force behind evolution, and can be split into different levels (individual-level, group-level, etc. selective forces). Please revise.

Answer: We agree that our text was incorrect. We revised our text according to the reviewer's suggestions to clarify that costly cooperation cannot be favored by selection acting at the individual level alone (lines 284-303). In fact, the entire discussion section was revised for clarity following all the reviewers' comments.

Related to this:

L15. "...swarming cooperation is favorable for multilevel selection..." It is the other way round. Multi-level selection can explain why swarming cooperation is favored. But, in the context of the above sentence the authors probably want to say "group-level selection could favor cooperative swarming because it causes population expansion."

Answer: We agree with the suggested change and revised the abstract accordingly (lines 14-15).

L16. "However, this strong group-level benefit does not invariably result in natural selection." Again, this reads odd. It is not group-level benefits that drive natural selection. Group-level benefits can drive the evolution of cooperative swarming. Again, in the context of the above sentence the authors probably want to say "However, this strong group-level benefit does not invariably result in cooperative swarming being favored by natural selection."

Answer: Here too we revised the text to accommodate the reviewers comment as well as the request from reviewer #1 to highlight the role of diminishing returns (lines 15-16). Thanks for the constructive criticism.

*2) As already pointed out in my previous review, I wish to see a more balanced discussion of how the authors' work relates to recent studies that have chosen similar approaches with other microbial cooperative traits. For example, key systems include fruiting body formation in *Dictyostelium discoideum* (reviewed in Strassmann & Queller 2011 PNAS) and *Myxococcus xanthus* (Smith et al. 2010 Science), as well as quorum sensing (Diggle et al. 2007 Nature), and siderophore secretion in *P. aeruginosa* (Griffin et al. 2004 Nature). Siderophore production is probably the best-studied microbial cooperative trait with studies having examined frequency and density effects (Ross-Gillespie et al. 2007 Am Nat, 2009 Evolution), the role of relatedness (Kummerli et al. 2009 Proc B), population expansion/ local competition (Kummerli et al. 2009 Evolution), and cost/benefit ratios (Kummerli et al. 2009 J. Evol. Biol). Referring to this body of work is essential to send a strong signal to microbiologists that group-level benefits are not sufficient for cooperative traits to be selected for.*

Answer: We acknowledge that the introduction and conclusion sections were indeed biased towards bacterial work, and lacked citations to important previous work. We have revised both sections substantially to correct this problem. We also now cite all the papers listed by the reviewer in the introduction (line 31).

3) L57-58. The question "Could there be other processes favoring costly swarming cooperation?" is quite vague. The authors could be more explicit by saying that their previous work focused on cost-reducing factors (i.e. prudent resource use), whereas here they focus on the role of relatedness, whereas costs are kept constant. A reference to Hamilton's rule would make this point even clearer.

Answer: We appreciate the reviewer's suggestion and we restructured the introduction section with Hamilton's rule. We explicitly state that our previous work (Xavier et al 2011, Mol Micro) analyzed cost-reducing factors whereas the present work focuses on relatedness while keeping costs constant (lines 52-63).

4) I found the result of an inverse bell curve in Fig. 3b quite interesting. I had expected to see a negative linear relationship. Why do cheats not perform best at lowest frequency? A more detailed discussion would be helpful.

Answer: The inverse bell curve is to be expected, given that the data shown represents the *change* in cooperator proportion as a function of the initial cooperator proportion. The study of Chuang et al (PMID:19131632) shows a similar inverse bell shape (See fig 2A in that paper, bottom panel). We added an extra sentence to explain this (line 128-130).

5) L215: remove "that"

Answer: The typo was corrected.

6) L269. In the context of this sentence, the reader might conclude that reference (3) is an example of wrong interpretation. This is of course not the case. Please reconsider phrasing. Instead, reference (8) is a clear case of an erroneous interpretation. This could be discussed in more detail, but is maybe not helpful here.

Answer: Indeed, it was not our intention to say that ref (3) (now citation #7) was an example of a wrong interpretation. We restructured the section to avoid confusion: "Recognizing that selection acts at multiple levels is key to analyzing microbial social traits (7) can help reveal novel mechanisms stabilizing cooperation (e.g. 21, 36, 49, 65, 66-70), and can eventually inspire therapeutic strategies against pathogens (38, 71, 72)"

Reviewer #3 :

SUMMARY OF THE MS

The authors address the question of the relative importance of group-level and individual-level selection in the maintenance of cooperation. They construct a new P. aeruginosa strain in which expression levels of biosurfactant are tuned via external concentrations of L-arabinose. In this new system they can control the cost to benefit ratio of the production of biosurfactant. They then study how the "individually costly" but "group beneficial" production of biosurfactant can be selected against invasion by non-producer individuals. To understand the outcome of two-strain competition experiments they propose a Price equation-based model. This enables them to separate group-level and individual-level selection acting on the competition outcome. They conclude that inducible producer is out-competed by non-producer under high mixing/low relatedness regime, and that high strain segregation is needed for fixation of inducible producer strain. Finally, they compare the tunable producer strain with a wild-type strain that doesn't get out-competed by the non-producer strain even under high mixing regime.

GENERAL COMMENTS

- The main message does not seem to be novel.

- Some results are novel but are not highlighted - specifically the last result on wild type phenotype out-competing non-producer strain.

- Introduction and conclusion give a biased vision of the field.

- Regarding technical aspects, the manuscript is convincing and solid. The engineered inducible producer strain is an useful tool and the authors provide an interesting comparison with the wild-type strain. Only one control is missing: at a fixed arabinose concentration, do engineered cells produce surfactants at a constant rate regardless of the cell density and the presence of non-producers, as expected?

Answer to general comments: We are happy that the reviewer finds this work to be convincing and solid. We have carried out new experiments to address the reviewer's suggestion. Specifically, we measure the surfactants produced using the anthrone assay (lines 117-125, and methods in 412-423) to compare the inducible strain and the wild-type and to assess the effect of a non-producer. The control experiment shows, as expected, that the inducible strain is not influenced by the presence of the non-producer. In fact, previous work from our lab using reporter fusions revealed no feedback of surfactants produced to the expression level of surfactant synthesis genes (PrhLAB promoter) – see van Ditmarsch and Xavier, BMC Microbiol. 2011.

MAJOR CONCERNS:

A major concern is that the main message reported by the authors is not very new and that the introductory background justifying its novelty is partially biased.

1. The introduction and conclusion provide a biased perspective of the literature in the field. 1) Some microbial systems such as Dictyostelium in which extensive mixing occurs have evolved a (very) costly cooperation strategy (Fortunato et al., 2003; Sathe et al., 2010). 2) More importantly, cooperation is not explained mainly through group selection in the field, it is rather the opposite; most of the studies propose individual-level selection and kin selection to explain cooperation in microbes (West, Griffin, Gardner, & Diggle, 2006). The background on which the authors base their study is thus not reflecting faithfully the current view on these questions.

Answer: Indeed, it is now clear to us that we overlooked citing several important contributions to the study of cooperation in microbes and that our text was biased. The introduction and discussion have been significantly improved.

We include the citations mentioned by the reviewer (e.g. line 31). Also, our intention was not to give a biased perspective of the literature. Indeed, there has been extensive work in the past decade to show that cooperation is not mainly explained through group selection. Our intention is rather to point out that even recently group-level selection arguments are sometimes naively used incorrectly, mostly in the microbiology literature. We now provide references (lines 24-25,292).

2. The novelty of the conclusions is not very obvious. The fact that a constant producer is outcompeted by a non-producer is expected based on previous theoretical studies and published experimental results on bacterial populations (Dao, Kessin, & Ennis, 2000; Doebeli & Hauert, 2005; Rainey & Rainey, 2003; Travisano & Velicer, 2004). The fact that high relatedness favors fixation of cooperators in these situations was also repeatedly shown (Griffin, West, & Buckling, 2004; Nadell, Foster, & Xavier, 2010; West et al., 2006).

Answer: We have highlighted the novelty of this work by explaining that this is the first time that high relatedness is shown to contribute to the evolution of swarming motility and, importantly, to compensate for an unnatural cost which is not present in the wild-type strain (lines 72-74).

3. However the data of Figure 7, illustrating the side-by-side comparison of the engineered and wild type strains, convey a more interesting message than the one highlighted by the authors. They find that a constant production, even at an optimal rate (best group-level fitness vs cost of the cooperator), is way outperformed by the wild-type production scheme. This was not directly predictable based on the previous paper describing the wild-type strain (Xavier, Kim, & Foster, 2011).

Answer: This is indeed an important point to highlight. Because one would expect a cost to modulating expression of certain genes, it could be that low constitutive production is a better

strategy than regulation. Here we show that it is not and we have expanded the discussion (lines 271-283, 314-319), and added new simulations to investigate the importance of the marginal benefit of wild-type (fig. S5, see also reply to suggested revisions point 2).

4. Results on growth driven cooperation and capacity driven cooperation are interesting but not sufficiently elaborated and are not at all compared to other literature in the area.

Answer: We have expanded our discussion in this section and added several new references to contextualize our findings (lines 328-354).

SUGGESTED REVISIONS

1. The parts of the introduction and conclusion on individual and group selection in microbes should be revised. This subject has been discussed in many previous studies and the view advocated by the authors is, in our opinion, not under-represented - if the authors think that it is the case, they should provide references.

Answer: Again, we agree that the previous manuscript may have presented a biased review of the field. We have addressed this point in the revision. See the answer above at Major Concerns #1.

2. It remains true that the maintainance of cooperation in natural systems is not fully understood. Experiments and theory indicate special conditions (structured environment, low mixing, high relatedness, differential adhesion) under which cooperators can outcompete defectors. On the other hand, in highly mixed natural populations these conditions are difficultly reached (because mixing is incompatible with high relatedness and structures environment). In this respect, the presence of a mechanism in wt strains for resisting the invasion of defectors (Figure 7) is interesting and would deserve more attention: is this mechanism not an essential factor for maintaining cooperation in this system?

Answer: We appreciate the reviewer's suggestion; this is indeed an important point to address in our manuscript. We have carried out new simulations and added a new supplementary figure (fig. S5) that illustrates the outcome of multi-level selection if the wild-type strain did not have an individual level selection advantage with regards to the defector strain. This new simulation shows that even in the absence of a mechanism that confers an individual-level advantage to the wild-type strain, the wild-type strain is still favored by multi-level selection. We have also included an explanation in the text from lines 276-282.

3. Growth vs capacity driven cooperation. This point would also deserve a more elaborate discussion and references to previous studies. Is it for instance related to the distinction between R and K-selection? Can we connect it to the model of <http://www.biomedcentral.com/1471-2148/12/61/>

Answer: Indeed, our discussion on growth vs carrying capacity driven cooperation can be related to r vs K-selection and we thank the reviewer from bringing up this point. We have expanded our discussion to include R vs K-selection and how it related to our conclusion about growth vs carrying capacity driven cooperation (lines 335-346). The paper mentioned above is a valuable theoretical contribution to the effort of showing that a cooperative trait that expands carrying capacity is likely to evolve despite the presence of defectors. Houchmandzadeh et al. provide an alternate approach that is complementary to ours and we have now included its citation in our discussion of growth vs carrying capacity cooperation together with other theoretical contributions (line 344).

In our opinion, the paper would benefit from bringing upfront these more original points, rather insisting on the predictable outcomes of competition between inducible producer and defector.

Answer: We agree that the original points are worth highlighting and we believe the revised version does this appropriately. Thank you for the suggestion.

MINOR CONCERNS

1. Figures are mis-numbered (There are 2 figures 3). Some data in the figures are not mentioned nor discussed in the text (Fig 3D). Please either include them in the text or leave them out of the manuscript. We apologize for the mis-numbering of the figures. This has now been corrected. Figure 3D is now discussed in line 216.

2. Some typos, including in the abstract.

Answer: Thank you, we have carefully revised the manuscript to correct any typos.

3. It would be very interesting to run simulations while changing the variance (partitioning) at successive rounds. How do various strains displaying different production regulations perform under such selection regimes? Such data might also provide outcomes that would not be easily predictable based on previous studies.

Answer: Indeed this is an interesting point. We assume, as often happens in social evolution studies, that subdivision of the global population into subpopulations has a constant variance. This could indeed change from one generation to the next but it is also possible that it would stay the same assuming there exists a mechanism that has evolved to generate this particular distribution. In any case, we expect that even if the variance of the distribution changed from one round to the next, the wild-type strain will be more robust as it can win in competition against defectors independently of how the global population is sampled. We have included this discussion in the manuscript in a new paragraph from lines 319-327.

4. The manuscript contains many anthropomorphic expressions that refer to *P. aeruginosa* cells, and in general microbe populations. Although these anthropomorphic expressions are common in many publications in the field, it may be best to avoid them to prevent any misleading interpretation based on their common-sense understanding. These terms include here "altruistic death", "prudent cooperation", "bacterial charity", "defector", "selfish individuals",...

Answer: Indeed, the terminology may be considered jargon that is unclear outside the field. Therefore, to avoid misleading interpretations, we have added a glossary with definitions for the terms as used in this paper (Table 1).

REFERENCES

- Dao, D. N., Kessin, R. H., & Ennis, H. L. (2000, July). Developmental cheating and the evolutionary biology of Dictyostelium and Myxococcus. Microbiology (Reading, England). Retrieved from <http://www.ncbi.nlm.nih.gov/pubmed/10878115>
- Doebeli, M., & Hauert, C. (2005). Models of cooperation Models of cooperation based on the Prisoner's Dilemma and the Snowdrift game. Ecology letters, 8, 748-766. doi:10.1111/j.1461-0248.2005.00773.x
- Fortunato, A., Strassmann, J.E., Santorelli, L., Queller, D.C. (2003) Co-occurrence in nature of different clones of the social amoeba, Dictyostelium discoideum. Mol Ecol 12: 1031-1038.
- Griffin, A. S., West, S. A., & Buckling, A. (2004). Cooperation and competition in pathogenic bacteria. Nature, 430(August). doi:10.1038/nature02802.1.
- Nadell, C. D., Foster, K. R., & Xavier, J. B. (2010). Emergence of spatial structure in cell groups and the evolution of cooperation. PLoS computational biology, 6(3), e1000716. doi:10.1371/journal.pcbi.1000716
- Rainey, P. B., & Rainey, K. (2003). Evolution of cooperation and conflict in experimental bacterial populations. Nature, 425(September), 72-74. doi:10.1038/nature01942.1.
- Sathe, S., Kaushik, S., Lalremruata, A., Aggarwal, R.K., Cavender, J.C., et al. (2010) Genetic heterogeneity in wild isolates of cellular slime mold social groups. Microb Ecol 60: 137-148. doi:10.1007/s00248-010-9635-4.
- Travisano, M., & Velicer, G. J. (2004). Strategies of microbial cheater control. Trends in microbiology, 12(2), 72-8. doi:10.1016/j.tim.2003.12.009

West, S. a, Griffin, A. S., Gardner, A., & Diggle, S. P. (2006). Social evolution theory for microorganisms. *Nature reviews. Microbiology*, 4(8), 597-607. doi:10.1038/nrmicro1461

Xavier, J. B., Kim, W., & Foster, K. R. (2011). A molecular mechanism that stabilizes cooperative secretions in *Pseudomonas aeruginosa*. *Molecular microbiology*, 79(1), 166-79. doi:10.1111/j.1365-2958.2010.07436.x

2nd Editorial Decision

18 July 2013

Thank you again for submitting your work to *Molecular Systems Biology*. We have now heard back from the referee who accepted to evaluate your revised manuscript. As you will see, the referee is now satisfied with the modifications made, and supports publication of the work.

Prior to formal acceptance of your manuscript we would like to kindly ask you to address the following points:

- The references need to be formatted according to the MSB guidelines.
- The Supplementary figure legends are currently in the main article file, they should be moved to the Supplementary Information.
- The Supplementary information needs to be combined into a single PDF file.
- Please provide a "standfirst text" and a "thumbnail image".

Thank you for submitting this paper to *Molecular Systems Biology*.

Reviewer #3 (Remarks to the Author):

The authors have revised their manuscript to follow the suggestions of the reviewers. In our opinion, the manuscript is now suitable for publication.

2nd Revision - authors' response

19 July 2013

Thank you for the second review of "Multilevel selection analysis of a microbial social trait". We are happy to see that the reviewers are completely satisfied with our revisions.

Reviewer #3 (Remarks to the Author):

The authors have revised their manuscript to follow the suggestions of the reviewers. In our opinion, the manuscript is now suitable for publication.

Answer: We are happy that the reviewer agrees that the paper is ready for publication.